# Approved Ambiguities: An Analysis of Applications for the Ethical Review of Animal Research in Sweden—Focusing on Harm, Benefit, and the 3Rs

**DOI:** 10.3390/ani15192771

**Published:** 2025-09-23

**Authors:** Svea Jörgensen, Elin M. Weber, Johan Lindsjö, Frida Lundmark Hedman, Helena Röcklinsberg

**Affiliations:** 1Department of Applied Animal Science and Welfare (THV), Swedish University of Agricultural Sciences, P.O. Box 7068, 750 07 Uppsala, Sweden; svea.jorgensen@slu.se (S.J.); johan.lindsjo@slu.se (J.L.); 2Department of Applied Animal Science and Welfare (THV), Swedish University of Agricultural Sciences, P.O. Box 234, 532 23 Skara, Sweden; elin.weber@slu.se (E.M.W.); frida.lundmark@slu.se (F.L.H.)

**Keywords:** harm–benefit analysis (HBA), 3R, Directive 2010/63/EU, Animal Ethics Committee (AEC), animal research, animal ethics, animal welfare

## Abstract

In this study, we investigate and discuss how animal research applications undergo ethical review in Sweden. We examine what information must be included in submitted applications according to law, how well real applications mirror these requirements, and how Animal Ethics Committees (AECs) handle said information. If the AECs cannot perform a harm–benefit analysis (HBA) or assess how the 3Rs are considered in the planned project, they cannot fulfil the legal requirements of an ethical review. By examining a selection of applications, we found that information about harm to the animals, project benefit, and adherence to the 3Rs (Replacement of animals, Reduction of numbers, Refinement of methods) was often insufficient or left out. This was partly, but not exclusively, due to the structure of the online application form. We highlight the importance of detailed, complete, and accurate information about harms, benefits, and the 3Rs in each application to facilitate a proper ethical review performed by the AEC. We suggest nine action points to improve the evaluation process—for the sake of applicants, AECs, and the animals used in research—and to ensure research quality, transparency, and public trust in the ethical review.

## 1. Introduction

### 1.1. Background

#### 1.1.1. Directive 2010/63/EU and Its Implementation Across Member States

A little over a decade ago, Directive 2010/63/EU of the European Parliament and of the Council of 22 September 2010 on the protection of animals used for scientific purposes [1] (henceforth referred to as Directive 2010/63/EU) saw the light of day. Directive 2010/63/EU acknowledges the intrinsic value of non-human animals (henceforth animals), raised the bar for the welfare of animals used in research across the EU, emphasized the importance of Replacement, Reduction, and Refinement (the 3Rs), and set the ambitious goal of the full elimination of animal use in research as soon as scientifically possible (Preamble 10). All member states were required to transpose the rules of Directive 2010/63/EU into national legislation by 2013, and to ensure that their country’s animal research thereafter abides by them. Directive 2010/63/EU specifies that all use of animals in research causing pain, suffering, distress, or harm equal to or exceeding a needle puncturing the skin (Chapter 1 Article 3 p. 1) requires ethical approval by a competent authority before being carried out. The ethical review must contain a so-called harm–benefit analysis (HBA) whereby the total harm inflicted on the animals is weighed against the predicted benefit of the project (Chapter IV Section 3 Article 38 p. 2d). It must also ensure project compliance with the concept of the 3Rs (Chapter IV Section 3 Article 38 p. 2b). Furthermore, the review process should be “comprehensive” (Preamble 38) and “transparent” (Chapter IV Section 3 Article 38 p. 4).

Member states have been free to fulfil the objectives of Directive 2010/63/EU by implementing its contents into national legislation as they see fit [2,3,4]. Countries that have previously established stricter animal welfare standards than the minimum levels set by Directive 2010/63/EU have been allowed to keep these, if they do not interfere with trade (Chapter I Article 2). Sweden is one such country. Hence, Sweden’s implementation of Directive 2010/63/EU in the form of The Swedish Board of Agriculture’s Regulations and General Advice on Laboratory Animals (currently SJVFS 2019:09, also known as the L150) [5], together with The Swedish Animal Welfare Act (SFS 2018:1192) [6] and The Swedish Animal Welfare Ordinance (SFS 2019:66) [7], contains, in many aspects, more extensive protection of the welfare for animals used in research than in many other member states. Furthermore, the threshold of what qualifies as animal research in Sweden is not only defined by the needle-prick criterion found in Directive 2010/63/EU (Chapter 1 Article 3 p. 1) but by purpose, i.e., whether the animal use in question is intended for scientific research or “other comparable” purposes. Consequently, the Swedish definition of animals used for research includes a broader range of animals, and Sweden compiles statistics according to both definitions [8].

Implementation of Directive 2010/63/EU has however not been without critique. A recent study has found discrepancies between its harmonization across countries [4], and the European Commission Expert Working Group concluded several years prior that the “significant differences” with which certain aspects of Directive 2010/63/EU have been incorporated into national regulations are “risking the main objectives of the Directive to deliver improved science and welfare and give a level playing field for the scientific community across the EU” [9]. In addition to this, the HBA, as the approved tool for ethical weighing, has been described as inherently flawed [10,11,12,13,14], and whether or not the 3Rs have been adequately considered by applying researchers may be difficult for AECs to assess [15,16,17,18].

#### 1.1.2. Ethical Review of Animal Research in Sweden

The role of competent authority in Sweden is shared by six regional Animal Ethics Committees (AECs) functioning as subunits of The Swedish Board of Agriculture (SBA). These committees are responsible for processing all applications for the ethical review of animal research. Rather than being affiliated with individual research institutions, they are distributed across the country’s major university cities, with meetings typically held at the local district court. To facilitate procedural compliance, all AECs are chaired and vice chaired by lawyers. Both must be deemed impartial by the appointing SBA and have experience serving as permanent judges (Chapter 7 Section 14 of the Swedish Animal Welfare Ordinance). The remaining twelve seats are equally divided between representatives of the scientific community (researchers, animal research technicians, animal research staff) and laypersons, thereby reflecting both scientific and public interests. Less than half of the laypersons may represent animal welfare organizations (Chapter 7 Section 14 of the Swedish Animal Welfare Ordinance). Furthermore, to enhance transparency and public insight, applications and AEC decisions are public documents and may be obtained by anyone upon request.

The application form used by all applying researchers is created by the SBA and is accessible through an online portal on the SBA website. Following submission through this portal, a decision must be made within 40 business days unless good reason is given for the delay (Chapter 7 Section 5 of the L150). Once the applications have been received, they are divided amongst so called preparatory groups (the AEC split into smaller units). If the application is found to be incomplete or incorrect, the applicant must be informed of this promptly and asked to provide the missing information (Chapter 7 Section 7 of the L150). According to the SBA, “[a]n application should be considered complete when the committee has received enough information to perform the ethical review” (see General Advice for Section 5 in Chapter 7 of the L150). Next, all applications, along with any additional information from the applicant or external experts, are distributed to the entire committee, together with the preparatory group’s recommendation for what the AEC should decide. Finally, the committee gathers at a plenary meeting and votes to approve (with or without special conditions), reject, or postpone the applications. Importantly, the AEC’s decision is independent of whether or not a project has already received funding. In other words, a project may still be refused ethical approval even if a funding body has deemed it scientifically valid enough to merit financial support.

#### 1.1.3. Our Research

In a recent pilot study, we found several issues within Swedish ethical reviews performed in 2017 [19]. Applications lacking relevant information relating to harm, benefit, and the 3Rs had been approved, prompting reflection on the reliability of the ethical approval process in Sweden. Dissimilarities between regulations and the digital application form were hypothesized as potential (at least partial) causes of the observed problems. To investigate whether or not these issues remain, a second more comprehensive and in-depth study has been conducted, to be presented in this paper.

### 1.2. Aims

The study has the following aims:i.To map which written information is required in the applications for the ethical review of animal research based on relevant regulations of the ethical review of animal research (Directive 2010/63/EU and the L150). Also, to map when said information should include a justification by the applying researcher.ii.To analyze how well (a selection of) applications from 2020 live up to legal requirements (see aim (i)).iii.To gain an overview of how well the application form used by applying researchers in 2020 corresponds to legal requirements (see aim (i)).

### 1.3. Structure of the Article

This paper will give an account of a study combining aspects of legal scholarship with empiric research and animal ethics-based reasoning and argumentation. The empiric part of the study contains a qualitative analysis of data ultimately converted to quantitative results.

In the upcoming sections, a recollection of how the material was gathered and analyzed will be presented. This includes a brief account of legal methodology and an explanation of its application in this study. Following this, the results section is divided into two parts: Regulatory Requirements provides an answer to aim (i) as an outcome of the legal methodology used, whereas Analysis of Applications for Ethical Review provides the answer to aim (ii). Outcomes of aim (iii) have not been assigned a separate results section but will be addressed where relevant throughout the ensuing in-depth discussion. Finally, suggestions for improvements to the review process and proposals for future research will be presented, followed by a conclusion summarizing the main findings and reflections.

## 2. Materials and Methods

### 2.1. Interpretation of Relevant Regulations

#### 2.1.1. Applicable Regulations and Legal Expertise

To ascertain the legal requirements put on the applicants and the AECs, respectively, the L150 (SJVFS 2019:9), Directive 2010/63/EU, and, where explicitly relevant, the Swedish Animal Welfare Act (SFS 2018:1192) and Ordinance (SFS 2019:66) were scrutinized. At the time of submission of this paper, the versions referred to herein remain in use unaltered. When needed, the Swedish Board of Agriculture (SBA)—holding preferential right of interpretation as the central competent authority responsible for formulating the implementation of Directive 2010/63/EU into Swedish regulations (as the L150)—was consulted.

#### 2.1.2. Legal Methodology

A description and systematization of current legislation, *de lege lata*, may be carried out using legal dogmatics [20] (p. 52). To investigate *lex ferenda* (“future law” or “the law as it should be”, also referred to as *de lege ferenda*) a legal analytical approach may be applied allowing for the use of a wider range of sources and greater freedom to use argumentation in support of one’s conclusions [20] (pp. 53–55). We chose this analytical approach as it may be used in law when looking to answer a specific question or emphasize a certain perspective, such as investigating how well a particular legal framework corresponds to its intended purpose [20] (p. 79).

To determine *lex ferenda*, we proceeded in accordance with a fixed set of steps. First, the regulations relevant for our study were interpreted according to the rule that national legislation in a member state is expected to reflect EU legislation. This follows from the hierarchal order of legal sources within the EU [20] (p. 48). In other words, the L150 was interpreted in the light of Directive 2010/63/EU. Then, if a hierarchy-based interpretation was not guidance enough or not easily applied, we moved to the second step. Here, we considered the task(s) of the committees and the aim(s) of the ethical review in accordance with legislative intent through teleological interpretation. To ascertain the underlying purpose of the regulations in question, preparatory works and doctrine were reviewed and the SBA consulted.

### 2.2. Analysis of Applications

The empirical part of the study presented in this paper builds on the aforementioned pilot study by Jörgensen et al. (2021) [19], with the addition of several methodological developments and adaptations. Most notably, the guide used for the analysis has been adapted to regulatory amendments made in the L150 since the time of the pilot. Subscores have also been added to the guide to reflect nuances within the data (see 2.2.2. Data analysis for details). Furthermore, the amount of data included for analysis has increased compared to Jörgensen et al. (2021) [19], from 18 sets of documents to 44.

#### 2.2.1. Data Collection

In Sweden, according to the principle of public access to information (*Offentlighetsprincipen*), documents created by or submitted to and stored by public authorities are, as a rule, classified as official documents (*allmänna handlingar*) and open to the public. To facilitate insight into the due process of the governmental exercise of power in accordance with The Public Access to Information and Secrecy Act (SFS 2009:400), such documents must be shared upon request. However, access may be denied or limited for secrecy reasons, such as to protect the confidentiality of individual persons or organizations.

For this study, multiple official documents were obtained, consisting of electronically submitted applications for ethical review in Sweden (including the non-technical project summaries, or NTSs) together with their corresponding decisions, preparatory group propositions, and any supplementary documents. None of the documents were subject to confidentiality restrictions. We plan to present our analysis of the decisions separately in a forthcoming paper.

In 2020, the total number of applications decided on by the AECs was 602. To determine a suitable sample frame, the diaries and protocols from 2020 (also official documents) were requested from each AEC. These were then cross-referenced to reveal the number of applications that had been (a) submitted, processed, and decided on by the AECs during 2020 and (b) classed as first-time projects and not mere requests of alterations to existing research. After this initial sorting process, a total of 420 applications and corresponding documents (105 from Stockholm, 23 from Umeå, 58 from Linköping, 67 from Gothenburg, 88 from Malmö/Lund, and 79 from Uppsala) were determined as eligible for inclusion. In a blinded process, 10% of each AEC’s total number of documents was then further selected through simple random sampling. By rounding up each number to the nearest whole digit, a total of 44 documents were ultimately selected. The documents were requested directly from the district courts responsible for maintaining the archives. In essence, all publicly available written documentation was requested, received, and analyzed.

#### 2.2.2. Data Analysis

The documents were analyzed using a customized form of deductive content analysis to determine how well their content corresponded to the legal requirements of the L150. This method of analyzing qualitative data follows a systematic approach, whereby predefined legal criteria serve as the theoretical basis for applying a structured guide to categorize and evaluate the content of the applications, rather than relying on an inductive exploration of the material [21,22,23]. Given the study’s focus on regulatory compliance, this deductive approach was particularly suitable as it ensured an objective and transparent evaluation based on established legal frameworks, rather than a subjective interpretation of the text [21,22,24].

Since there exist no guidance documents for how researchers should fill out the application, the theoretical basis for the analysis—i.e., the legal criteria—was established through legal interpretation (previously described under 2.1.2.). Based on this interpretation, a guide for the subsequent content analysis was developed in advance to ensure transparency, reduce judgment bias, enhance internal and external consistency, and enable reliable reproducibility. While this guide does not represent how all parties comply with regulatory requirements, it provides a thorough and systematic approach to our interpretation of the L150 and Directive 2010/63/EU, integrating legal methodology along with clarifications and corrections from the SBA where relevant. The full guide is available under the Appendix A.

The content analysis focused mainly on identifying whether the applications met regulatory requirements regarding basic information (e.g., species, origin, and developmental stage of animals), harm (to the animals), benefit (of the project), and the 3Rs. To ensure that information about harm or benefit was not overlooked, the applicants’ own ethical deliberation (provided in a designated section of the application form) was included in the analysis, despite not being legally required content. Due to how the regulatory requirements of the L150 are structured, we analyzed the main technical part of the application form and the non-technical project summary as separate entities.

The examined contents were scored depending on how well they fulfilled the set requirements (see Appendix A). Most content was assigned one of three scores: Y (Yes, information provided), I (Incomplete/Insufficient/Indeterminable information provided), or N (No, information not provided). In some cases, subscores (y and Y!) were used for additional detail. For some information, certain scores were not applicable (N/A). Two researchers conducted the analysis independently, blinded to each other’s results. Upon completion, the results were compared, and any discrepancies were discussed, analyzed, and decided upon anew in unison by the two researchers.

Finally, descriptive statistics were used to summarize the extent to which the applications complied with legal requirements. By quantifying the categorized data, trends in completeness could be systematically assessed, and common gaps in the submitted applications were identified.

## 3. Results

### 3.1. Regulatory Requirements

In the sections below, we present regulatory requirements concerning the written documents of the ethical review process according to the hierarchal order of legal sources. Consequently, the EU Directive is addressed first, followed by the Swedish national legislation, i.e., the L150. Whenever cross-referencing of the two allowed for more than one conclusion, the SBA was consulted, and legal methodology was applied as previously described under *Methods.*

#### 3.1.1. Content of Applications

The information to be included in the application for ethical approval is specified in Article 37, Article 38, and Annex VI of Directive 2010/63/EU. Article 37 states that an ethical application “shall include at least” a project proposal, a non-technical project summary, and the elements set out in Annex VI (Figure 1):

This describes, in other words, the *minimum amount* of information an applicant must provide when applying for the ethical review of a project. The European Commission Expert Working Group states the following: “To facilitate the harm-benefit analysis, sufficient information must be included in the application, to enable the evaluators to make a reasoned judgement on the harms to animals and the benefits likely to accrue from the project and the likelihood of these being achieved.” [25]. This statement is mirrored by Article 38 describing the ethical review process (with the exception of the likelihood of success, which is not mentioned). What “sufficient information” might be, if surplus to the points listed in Article 37 and Annex VI, is not exemplified, nor is there any elaboration on how the term “sufficient” is defined in this context.

In the L150, a corresponding provision can be found in Chapter 2 Section 15 describing the electronic application form for ethical approval, and Section 16, outlining the contents of the non-technical project summary (the NTS). Chapter 2 Section 15 is worded, in its English translation provided by the SBA for the Nordic Consortium for Laboratory Animal Science Education and Training (NCLASET) [5], as follows (Figure 2):

Given the ambiguity of the term “relevant” in the introductory sentence of Chapter 2 Section 15 of the L150, we contacted the SBA to ascertain how to interpret it correctly. They explained that the term “relevant” can be interpreted from either the perspective of the AECs or the applying researchers. The 13 points should as a rule be included in the application but, depending on the research project, any one could be deemed irrelevant and left out (SBA Research Animal Unit, personal communication, 7 December 2022). The SBA further stated that it is up to each individual AEC to decide, on a case-by-case basis, what information is necessary to reach a decision. Furthermore, the SBA described that “relevant*”* may also be seen as guidance for the applicant to ensure that they only provide information of importance for the AEC’s deliberations. If information has been excluded, this could hence be seen as a conscious choice by the applicant due to issues not being applicable, or because the applicant regards the information in question irrelevant for the decision-making process (SBA Research Animal Unit, personal communication, 7 December 2022).

#### 3.1.2. Motivation of Content of Applications

Annex VI of Directive 2010/63/EU specifies that “relevance and justification” should be given for the following: (1a) the use of animals including the origin, estimated numbers, species, and life stages, and (1b) procedures. The ensuing points stand on their own and contain no mention of relevance and/or justification.

Chapter 2 Section 15 of the L150 states that “motivated information” should be included in the application “where applicable”, which, interpreted in the light of the superior Directive, becomes relevant for at least the points requiring “relevance and justification” in Annex VI. (It is important to note here that the word “motivated” used in the English version of the L150 is a direct translation of the Swedish term *motiverad*, which in Swedish more accurately means “justified” or “rationalised”. In line with this, in Annex VI of Directive 2010/63/EU referenced above, the phrase “relevance and justification” has been translated to “relevans och motivering” in the Swedish version of the text. Thus, in this context, “motivated information” as specified in the L150 is to be understood as “justified information,” rather than in the everyday English sense of being driven by personal motivation or incentive.)

When “where applicable” in the context above is analyzed using legislative intent, a motivation would be required if its inclusion would be helpful for the AEC in forming a well-thought-out decision. According to the SBA, this means that AECs may require motivation for information relating to any of the 13 points on the list or any additional information they require, if they deem this information necessary to include in the first place (SBA Research Animal Unit, personal communication, 7 December 2022).

### 3.2. Analysis of Applications for Ethical Review

Out of the 44 sets of documents obtained for analysis, 11 were from Stockholm, 3 from Umeå, 6 were from Linköping, 7 were from Gothenburg, 9 were from Malmö/Lund, and 8 were from Uppsala. All applications requested, and were granted, approval for the longest available period of five years. The number of animals to be used in each study ranged from 10 individuals to 13,600. Five applications included, at least in part, projects on wild animals and four included privately owned animals.

In the upcoming section, the results have been grouped thematically according to the regulatory requirements that they correspond to, i.e., if they relate to basic information about the project or animals used, harm to the animals, benefit and purpose, or the 3Rs.

#### 3.2.1. Basic Information

Basic information about the animals was not always provided to the full extent by the applying researchers (Figure 3). The majority of applications justified and described the relevance of using animals of a certain species, whilst only two did so for the animals’ developmental stage. More than half of the analyzed applications contained no information at all about the developmental stage (even if relevant), and almost one third did not include any information about the origin of the animals.

#### 3.2.2. Harm and Suffering

Animal harm or suffering was commonly only partially reported by the researchers (Figure 4). To obtain the highest score for this topic, either within the applicant’s “own ethical weighing” or within the NTS, some mention of the intensity/frequency and duration of the proposed harm had to be made, even if brief. For reference, one of the three applications, which lived up to this requirement within the section “own ethical weighing”, did so by explaining that “… [t]hese trials are of relatively short nature and only involve limited interventions to the animals” (our translation), where “short” was understood as duration and “limited” as describing intensity. Almost 1/3 and 1/5 of applicants made no mention of the harm at all within the main technical part of the application form and the NTS, respectively.

It was common for the NTSs to contain descriptions of planned procedures but without mentioning how they would impact the animals or possibly cause physical or mental suffering. Many descriptions of procedures were simplified to the point where important information was completely left out. As a result, the project may have appeared less harmful in the NTS than could be understood from the main technical part of the application form. In several cases, the NTS only included a selection of the planned procedures, or all were described as one. To illustrate, one NTS stated that “Some of the animals will develop symptoms at parity with the diseases we are studying” (our translation) as the only mention of what the experiment would mean for the animals. Yet another claimed that there would be no negative effects for the animals, despite a number of invasive procedures having been listed in the main technical part of the application form and the project being categorized by the AEC as severe. The one NTS graded as Y provided an account of suffering specific for the project with a mention of the intensity/frequency and duration: “The animals are mainly subjected to simple procedures, but these are repeated or may be combined. The animals will be affected for a shorter period and may show signs of mild discomfort. The application does however also contain two inflammation/infection models where the mice used will become sick from the treatment.” (our translation).

Scientific end-points (i.e., when the scientific aims have been reached) were provided for almost all studies, whilst only five applications described humane end-points (i.e., physiological or behavioral indicators that define the point at which an animal’s pain and/or distress must be reduced or terminated) for all planned procedures accompanied by relevant and clear assessment criteria (Figure 4). A proposed degree of severity was provided by the applying researchers for all projects. Whether these degrees correspond to the final degrees decided upon by the AECs has not been investigated.

We initially analyzed how the animals were to be housed and cared for *before*, *during*, and *after* their use, as specified in the L150 (Chapter 2 Section 15 p. 11). However, the scores are not included as they were found to be misleading, partly due to the application form not providing designated space for this level of detail, but also as a result of this information not being equally possible to provide for all kinds of projects. Similarly, information regarding methods of euthanasia will not be presented as on multiple occasions, it was uncertain if an application unintentionally omitted this information for some animals or if said individuals were in fact not destined for euthanasia at all.

#### 3.2.3. Benefit and Purpose

Compared to harm and suffering (Figure 4), benefit(s) of the research project were more often thoroughly described by the applicants (Figure 5). The majority of applications included information in both the main technical part and the NTS about what and/or how great the benefit of the project was expected to be and were as such scored as Y. In order to obtain the highest score Y!, the likelihood of achieving said benefits also had to be included. Only one applicant made any mention of the likelihood and did so within their “own ethical weighing” by claiming that their study had “high translational value” and “great potential of fulfilling its aims” (our translation). Several applications contained statements along the lines of “our project will result in x”. Such expressions of absolute certainty by the researcher were not regarded as descriptions of likelihood, as they lacked any supporting estimations or calculations of achieving the predicted outcome.

Almost all applications accounted for the purpose of the planned project (Figure 6). However, some applicants described the purpose of their research when asked about the benefit, or vice versa. This mix-up was present in both the main technical part of the application form and in the NTS.

#### 3.2.4. Replace, Reduce, and Refine (The 3Rs)

There was a great spread amongst applications regarding how well the 3Rs were described (Figure 7). In the main technical part of the application form, as well as the NTS, Refine was most often only partially described (I). Reduce stood out as the R least often mentioned at all within the main technical part of the application, whilst Refine was the least mentioned in the NTSs. During the analysis, it was difficult to determine where to draw the line between Refinement and standard practice as both may differ among institutions, projects, and practitioners. Moreover, within the scope of this study, it was not possible to compile a sufficiently comprehensive list of experimental methods and corresponding Refinement measures to serve as a meaningful benchmark for this distinction. For this reason, the applicants were given the benefit of the doubt, and strategies or actions described as Refinement were accepted as such, unless clearly false. This meant that for example “keeping fish in a tub of water whilst they recover” was graded as Y although it was not entirely certain that this was not simply common practice or what a reasonable alternative course of action would have been. Scenarios where it was clear that the proposed action was in fact not a form of Refinement were however not accepted as such. For example, one applicant described the perfusion fixation of mice as refined by anaesthetizing the animals beforehand, when, in fact, the procedure may never, under any circumstances, be performed on conscious animals.

In 15 of the NTSs, at least one of the 3Rs was confused with and inaccurately described as another. The most common mix-up was between Reduction and Refinement. In the main technical part of the application, the frequency of this occurring could not be determined since descriptions of how the 3Rs are applied to the project were not explicitly requested therein by the application form in use at the time. Hence, it was difficult to determine which R the applicant described and/or if the researchers knew which one they were referring to in these cases.

## 4. Discussion

### 4.1. Ambiguities Concerning Regulatory Requirements

The first aim of this study was to clarify content requirements for applications for ethical review based on an overview of the relevant regulations. As shown, Directive 2010/63/EU specifies relevance and justification to be provided only for the first two points in the list in Annex VI. In the L150, however, the phrase “relevant, and where applicable, motivated information” has been placed to apply to *all* subsequent points, allowing for a greater flexibility for the AECs to decide themselves which information is necessary (SBA Research Animal Unit, personal communication, 7 December 2022). On the one hand, it allows for the possibility that more points are justified, i.e., goes beyond the requirements of Directive 2010/63/EU. On the other hand, we see a risk that said flexibility would contradict Article 37 of Directive 2010/63/EU if the information listed in Annex VI is left out.

The above highlights an inherent challenge in manifesting a harmonized legislation across EU member states whilst simultaneously allowing for variations in its interpretation through subjectivity and case-by-case decision-making [4,9,15]. We argue that allowing the scope of “relevance” to be decided on an individual basis as described by the SBA may be precarious for several reasons.

First, the presence of vague regulatory formulations leaves room for interpretation [26,27,28,29], and when coupled with a lack of access to guidance documents [30], this risks creating uncertainty of what information the ethical review should be based on. In practice, this could make it difficult for researchers to predict what they will be asked to provide and consequently allow for a great variation in the information provided. It would also cause insecurity amongst regional AECs and individual committee members of what information or level of detail they should be receiving and requesting. Second, it sets high expectations for the collective knowledge of each AEC regarding research fields, the physiology and biological needs of specific species, available methods and experimental design options, recent developments in Refinement, etc., to allow for an accurate assessment of what might be “relevant” information for each project. This knowledge is not always present [31,32]. Third, we see a risk that this flexibility may undermine the value of important core principles and intentions manifested in Directive 2010/63/EU, as well as uniformity between EU countries.

### 4.2. Analysis of Applications by Researchers

The second and third aims of this study were to investigate how applications for ethical review from 2020 fulfilled the regulatory requirements (aim (ii)) and how well the application form coheres with those same requirements (aim (iii)). The following sections will provide a discussion of the outcomes of the results related to both of these aims.

#### 4.2.1. Basic Information

As shown, much of the basic information was insufficient or entirely missing amongst the analyzed applications. Part of the reason for this could be the lack of clarity of how the legal framework should be understood and applied. Another reason might be that the application form itself does not promote inclusion or emphasize the importance of the information in question. For example, the application form only asks the researcher to state the age of the animals if they consider it relevant. This directly contradicts Annex VI p. 1a of Directive 2010/63/EU. However, it mirrors the flexibility derived from the first sentence of Chapter 2 Section 15 of the L150, despite the repeated use of the word “shall” therein, for example as in point 6: “The number of laboratory animals and their origin as well as their developmental stage *shall* also be specified here.” (our emphasis), potentially facilitating the perception of an obligation to provide certain information rather than a mere opportunity to do so.

It is worth noting that some information may be inherently difficult for researchers to know, and thus share, depending on the type of project that they are planning on conducting. For wild or privately owned animals, origins may be uncertain. Likewise, the sex, age, number of animals, or even species may not be foreseen for randomly selected animals. In our study, two applications concerned the use of animals for the training of medical/veterinary students and staff, and neither specified the total number of animals to be used (nor what they would in fact be subjected to). Yet another two applications concerned the catching of wild fish for population inventory purposes. Both provided an estimate of the number of fish to be caught, but neither could confirm the accuracy of these figures, attributing the uncertainty either to the fishing methods used or to the unavoidability of bycatch. For the same reasons, the specific species of fish (or other aquatic animals) to be caught could not be determined beforehand. Although it is understandable that researchers planning projects outside the norm of using destination bred animals within the confines of a lab may struggle to provide certain details [33], the importance of this information for an accurate assessment of total harm may in some cases make its absence troublesome. Knowing an animal’s origin, developmental stage, or species may be vital in order to accurately assess how it is likely to experience and cope with stress, anxiety, or pain [34,35,36,37,38,39]. In essence, the importance of basic information should not be overlooked, as even the smallest pieces are needed to complete the whole puzzle.

In the spring of 2023, during the course of this project, the SBA released an updated version of the digital application form. Like the previous version (used for the applications analyzed in our study), the new form is not accompanied by a guidance document. Although a complete overview of all revisions has not been performed within the scope of this study, we are aware of some alterations that we believe are clear improvements. For example, the new form includes labels indicating to the applicant what information they *must* or merely *may* provide. We have not investigated to what extent these labels accurately reflect regulatory requirements, but it is our expectation that the new form may facilitate both researchers submitting applications and AEC members assessing them. A thorough investigation of how the new form is perceived by applying researchers and AEC members and how it may have affected the ethical review process would be interesting to perform once it has been in use for longer. Importantly though, even after revision, there is still only one available version of the application form. The European Commission Expert Working Group has clearly stated that different application forms for different projects may be necessary [25], and, as this study has shown, some requested information may be inherently impossible to provide using a “one-size-fits-all” model. Consequentially, even if the new form represents a step in the right direction, it may still fall short in supporting the submission of all information required by legislation across different research fields and project types.

#### 4.2.2. Harm and Suffering

To perform an HBA, one must first gain an overview of benefits and harm. Regarding the latter, predicting the cumulative harm is crucial. Cumulative harm consists of all negative impacts on an animal’s life whereby longer exposure equals more total harm caused, whilst the repetition of procedures may either increase the harm (sensitization), allow for acclimatization, or even reduce sensitivity to stress, etc., through habituation [34,40,41,42]. To assess the cumulative effect of harm, one must therefore know not only what kind of harm will take place, but also other factors, such as its intensity, frequency, and duration [43], which, in addition, may be affected by the animal’s housing conditions, previous experiences, and expectations.

Remarkably, there exists no explicit regulatory requirement in neither Annex VI of Directive 2010/63/EU nor Chapter 2 Section 15 of the L150 that harm should be described on its own within the main technical part of the application form. However, the European Commission Expert Working Group has stated that procedures “should be described in sufficient detail to enable harms to be assessed” [25]. In line with this, both the old and the new versions of the Swedish application form require a detailed description of interventions and how these will affect the animals. As seen in our study, provided information can vary greatly between applications. Thus, ethical committees may have, to varying degrees, to read between the lines to deduce suffering. If an applicant has provided clear and comprehensive descriptions of standard procedures where the impact on animals is well-known, estimating the expected suffering on this basis may be possible. For novel procedures, however, or when animal responses are difficult to predict, this might be less straightforward. It may also be challenging to estimate harms if the application itself is too extensive to allow for a good understanding of what is to be done to which animals, a phenomenon not uncommon amongst the applications analyzed in our study. On the one hand, asking AECs to deduce cumulative harm to the animals in this way may reduce the risk that they too readily adopt the researcher’s prediction of harm. On the other hand, the AECs in their current composition may not always possess the necessary spread and depth in knowledge to accurately assess suffering for all kinds of animals in all possible situations within all fields of research.

Both Directive 2010/63/EU and the L150 require the applicant to account for harm within the NTS. The application form also requests the applicant to provide their own so called “ethical weighing” similar to the HBA performed by the AECs. In spite of this, the standard dimensions intensity, frequency, and duration of the proposed harm are not specifically requested for either of the documents. Consequently, this level of detail was provided by only a handful of applicants. Moreover, even though the NTS should be “accurate and representative of the project” [44], deficient project or procedure descriptions, diminishing the associated harm, were not uncommon in our study. In such cases, laypersons of the AECs who may be reliant on the NTS will not be provided with the full picture of the animals’ situation. For obvious reasons, this is cause for concern. Furthermore, as the NTSs are available for anyone to read by visiting the ALURES Non-Technical Summary EU Database [45], inaccuracies therein would, if unidentified, lead to the general public falsely perceiving experiments as less harmful than what they actually are, whilst, if discovered, likely harm public trust in animal research. In either scenario, transparency is obstructed.

For animals used in research, simply living under laboratory conditions may have a negative impact on their health and wellbeing [46]. According to Directive 2010/63/EU (Annex VI p. 10), the general housing, husbandry, and care conditions of each project must be detailed in the written application, and this should be performed with the same level of detail across applications [25]. However, what constitutes *general* is not specified, and it is unclear what level of detail is deemed appropriate for this kind of information. Is it enough to say, as seen amongst the documents we analyzed, that “cages live up to EU standards”, “animals will be handled by trained staff”, or that “mice will be group housed for the most part”? Or should AECs be allowed (or expected) to request detailed cage blueprints, staff certificates, and exhaustive lists of enrichment options and animal conflict-prevention strategies? After all, it makes quite the difference to both the animals and the scientific validity of the project [47] whether or not rodent cages enable the performance of natural behaviors [48,49], mice are handled in a stress-free manner [50,51,52], and active measures are in place to enable the harmonious group housing of social animals [53,54], etc.

The L150 goes further than Directive 2010/63/EU, by requesting “[a]n account of how the laboratory animals are to be kept and cared for *before*, *during* and, if relevant, *after* the animal experiment” (Chapter 2 Section 15 p. 11, our emphasis). However, this level of detail was not requested by the application form, and the space provided therein to describe housing and care was limited. In fact, despite constituting a substantial part of the animals’ overall experience, daily care routines lack an allotted space in the form altogether and, as a result, are rarely described at all. Only if researchers wish to deviate from regulatory minimum requirements are they asked to elaborate and justify their wish to do so. Specific care regimens (e.g., extra observation, administration of analgesia, or altered feeding regimens) that the researcher may wish to apply to animals with certain needs (for example caused by invasive procedures or progressed illness) may be elaborated on in direct relation to where procedures are described. As a general rule, information about housing comes as a pre-written body of text automatically included in the application form based on which establishment the researcher is affiliated with when they log in to the system. Amongst analyzed applications, this information was often short, vague, and in some cases entirely irrelevant for the project or species in question. Applying researchers can edit or add to this body of text, but this is seldom done and, if so, rarely applies to all animals used in the project.

Why the application form encourages less detail than the L150 requests is unclear. It is clear, however, that fulfilling the requirements of the L150 may not be equally manageable for all researchers. For instance, if privately owned animals are used, their “before” may not be known, and if experiments are terminal, it is unclear what one should define as “during”. Likewise, for animals who are to be released back into the wild, returned to their owners, or rehomed, their “after” is impossible to ascertain. Although arguably helpful for assessing cumulative harm and gaining an understanding of the lives of the animals, this level of detail could make already complex and lengthy applications even more difficult to grasp, especially for laypersons, and the AECs, already bound by time constraints, would likely struggle to find the time to gather and process this information [55]. Amongst the applications analyzed in our study, apart from it frequently being unclear *which* animals were to be kept or cared for *how*, it was often difficult to decipher *when* aspects of housing or care would take place. As a result, we did not include an analysis of this in our final dataset.

Approvals granted without AECs having a full understanding of how the animals are to be kept, cared for, and potentially harmed indicate that it is impossible to determine whether appropriate Refinement measures have been applied. It also indicates that such approvals are not based on comprehensive HBAs. This could jeopardize animal welfare in a number of ways. For example, AECs may, as a result, unintentionally approve inappropriate housing conditions or miss opportunities for Refinement. A worst case scenario would be the granting of ethical approvals for projects where the legal “upper limit” of permitted suffering (see Article 15 p. 2 of Directive 2010/63/EU) is at risk of transgression due to incorrect severity assessments, perhaps in turn based on an inadequate understanding of the cumulative effects of multiple harms.

#### 4.2.3. Benefit and Purpose

Benefits of a project are generally considered more difficult to assess, quantify, and predict than harms to the animals [56,57]. Despite this, Directive 2010/63/EU provides little advice on how to assess them [58] whilst, as reflected in the L150, specifically requesting that they be described within both the main technical part of the application form and in the NTS.

Most applicants in our study include information about the benefit of their project, while excluding an account of the likelihood of achieving said benefits. This might be a direct result of neither the L150 nor the application form requesting this information. However, an estimate of likelihood constitutes a vital piece of information for a proper HBA [25,59,60,61] and the European Commission Expert Working Group motions for the application form to “invite these questions to be addressed” [25]. For this reason, it was included in our analysis (see Appendix A for details).

Instead of providing calculations or estimations of likelihood of success, many researchers describe the success of their research as an absolute certainty, e.g., in terms of “our project *will* result in…” or “we *will* achieve…” (our emphasis). Such predictions are, however, known to be uncertain [57], and recent figures show that, in hindsight, only 59% of questioned researchers in Sweden believe that their research did in fact achieve its goals [62]. The AECs seemingly seldom challenge statements of absolute certainty or ask to know how the likelihood has been estimated. This may indicate that one or more of the following situations apply: that AECs are inclined to trust the word of the applicant, which the European Commission Expert Working Group directly advises against [25]; a lack of sufficient time to thoroughly prepare each application [63]; or that a study may be perceived as ethically robust simply due to its approval by a funding body. Further, the AECs may be inclined to automatically assume that benefits will be substantial, regardless of the envisaged outcome of the work conducted, if the research field itself holds a certain status [64].

Whatever the reason, including benefits in the HBA without an estimate of the likelihood that they will be achieved makes it impossible to perform a proper HBA [65]. To estimate the probability that a project will deliver what it promises, AECs should therefore pay special attention to, for example, the choice of methods, proposed time frames, and how research questions have been formulated. However, we argue that there are good reasons to demand even more than Bout and co-workers’ threshold of likelihood, by considering the *quality of research* (one of the three dimensions in the Bateson cube) as a first and fundamental requirement for a proper ethical evaluation. Only when reflections of the validity ensure good research practice and the research design safeguards against pitfalls in terms of validity, e.g., through application of the 3V-approach proposed by Würbel and Eggel [66,67], should the next step of estimating the importance of the research be performed, followed by a weighing of harms against benefits [61]. Extending the ethical evaluation to also include an assessment of internal, external, and construct validity, could maximize scientific validity overall and, by extension, increase both the likelihood that the benefits of approved projects are achieved and that projects with low validity are not approved. That is, research proposals lacking scientific quality will be less likely to lead to expected knowledge or benefits and should warrant vigilance. Otherwise, approvals by the AECs not only risk permitting unnecessary use and suffering, as well as low-quality research, but could also fuel an already existing critique that ethical approvals are granted based on implicit promises rather than explicit scientific soundness [17,32]. It may also give the impression that AECs act less like impartial review boards and more like caterers to the needs of animal researchers at the cost of animals [15].

Further, it has been widely discussed whether new scientific knowledge achieved through research should per se be considered a benefit. We agree with Grimm and Eggel [10,68] that knowledge may indeed be beneficial in itself and that the tendency to rank this lower than applied benefits could be questioned. If not, researchers may resort to speculations and exaggerations of potential benefits in an attempt to secure ethical approval [10]. They may also, as observed in our study, describe general benefits attributed to the wider research field rather than the expected outcome of their particular project. To prevent this and to facilitate a review of basic research, Eggel and Grimm suggest a shift from a harm–benefit analysis to a “harm–knowledge” analysis. This would also simplify the work of the AECs by delegating the question of benefit to a retrospective assessment, thus limiting the scope of their evaluation [68]. Asking researchers to predict the future in the shape of benefits is asking a lot and could, we believe, undermine the intended function of the HBA if AECs are unable to ascertain when said predictions do or do not accurately reflect reality. We do however regard it comparably problematic to assess knowledge (as benefit), not least since its achievement is difficult to measure, but also because it is equally unsuitable as a counter-weight to harm. Additionally, all research, even failed projects, could be said to generate knowledge (of sorts). Consequentially, when weighing harm against knowledge, it may be difficult to find a balance between enabling important basic research for which tangible benefits are not easily formulated and risking the approval of insufficiently planned projects, which may cause harm for no greater reason than general inquisitiveness. Moreover, a systematic retrospective assessment of benefits in relation to harms would cause substantial time lags in the ethical evaluation [61], offering only a utilitarian verdict of whether or not the harm caused was ultimately justified. This would undermine the protective purpose of the HBA, which is to determine *in advance* which harmful actions should be prevented.

At the same time, the declaration of highly specific benefits or guaranteed success in the application could raise doubt about the need to conduct said research at all if the outcome is already known. In fact, Directive 2010/63/EU states that unjustified duplications of procedures must be avoided where appropriate (Article 46 Annex VI p. 9). In line with this, Eggel and Grimm state that research that does result in something previously known would be illegal to perform [68]. Hence, caution is warranted by the AEC when reviewing these kinds of projects and they should take care to evaluate both the kind of benefits expected (applied benefits or scientific knowledge), as well as the likelihood of success, and, ideally, the project’s validity.

#### 4.2.4. Replace, Reduce, and Refine

According to available regulations, the 3Rs should be described within the main technical part of the application (see Article 38 p. 2b of Directive 2010/63/EU, and Chapter 2 Section 15 p. 7 of the L150), as well as within the accompanying NTS (Article 43 p. 1b of Directive 2010/63/EU, Chapter 2 Section 16 p. 4 of the L150). The Swedish application form instructs the applicant to describe within the main technical part of the application form why animals must be used and what alternative methods have been considered, how the number of animals have been calculated and minimized, and what will be done to reduce the animals’ suffering for each planned procedure. However, neither of the terms Replacement, Reduction, nor Refinement were specifically mentioned in the main technical part of the application form used in 2020, and we believe that this may have made it unclear to the applying researchers what they were in fact being asked and why and that this may be a reason they so often did not provide this information. In the new form from 2023, all three terms are included. In the NTS, however, it was explicitly stated, also in 2020, that information about each R should be provided. However, amongst the NTS analyzed in our study, the 3Rs were, to varying degrees, insufficiently or not at all described, and one third of the NTSs displayed some mix-up between them. Despite this, there were no notes amongst the analyzed documents indicating that the AEC in question confronted the researcher about this noticeable confusion or discussed it internally. In one application, the 3Rs were repeatedly confused with one another, both within the original application and in replies to questions sent to the applicant by the committee. Nevertheless, the committee included no written mention of this and approved the application. A lack of understanding of the 3Rs, as well as confusion between them, is not a new phenomenon. In fact, several studies have described this at various levels among researchers, AECs, and Animal Welfare Bodies [18,69,70,71,72].

Distinguishing among minimum requirements, standard practice, and Refinement strategies is not always easy. Also, minimum legal requirements do not necessarily equate to minimum ethical standards [73], and researchers and AECs may thereby hold disparate views of when certain actions, such as pain relief or gentle handling, fulfil a bare minimum or is a form of Refinement. This could of course influence the extent to which researchers provide information about the 3Rs in their applications and make it difficult for AECs to question researchers’ supposed Refinement plans even if they may feel that more could and should be done. Descriptions of both Refinement and methods used are elements that should not vary in the level of detail between applications [25], yet our results show that this is in fact precisely what occurs in practice. This implies a substantial risk to the envisaged harmonization of research conditions within the EU and hence a need to highlight the found discrepancies between requirements in legislation and requests in the application form.

When AECs are expected to base their decisions on more information than researchers provide them with, a very complicated and ultimately problematic situation emerges. The main concern here lies in the simple fact that without an understanding of *how* and *to what extent* the harmful impacts on an animal’s life have or have not been mitigated, it is impossible to calculate cumulative harm, and as previously discussed, without knowledge of cumulative harm, a proper HBA cannot be made and an ethical review may not be lawfully completed. Likewise, without proper knowledge about Replacement strategies, the ultimate goal in Directive 2010/63/EU of the complete elimination of animal use in research is hampered.

#### 4.2.5. Limitations of the Study

In our analysis, we did not perform any subjective estimations of the magnitude of harm based on descriptions of procedural methodology as it was important that each document was assessed on equal terms and that the analysis could be repeated. This is, as such, a known limitation of our study, and we cannot rule out that it has likely, albeit to an unknown extent, been done by the AECs (described under 4.2.2 as reading between the lines), although no annotations thereof were found in any of the analyzed decisions.

## 5. Suggestions for Improvements

Based on our detailed examination of the challenges and shortcomings within the Swedish ethical review process of animal research, several suggestions for future actions and improvements emerge:

### 5.1. Creation of Comprehensive Guidelines Clarifying Regulatory Requirements:

Develop clear, detailed guidelines focusing on regulatory requirements for researchers and AEC members to decrease interpretation ambiguities and align the understanding of tasks and responsibilities. To ensure harmonization across member states, seek advice from the EU Commission on how to formulate said guidelines.

### 5.2. Revision and Improvement of Application Form(s):

Evaluate the impact of the recent update of the application form on the Swedish ethical review process and identify further areas for improvement. Continue to update and improve the application form, ensuring that it aligns with regulatory requirements and aids researchers in providing the necessary information. Consider creating a range of application forms tailored to different types of research projects, addressing the unique needs and challenges of each, to facilitate a more effective review process.

### 5.3. Revision of AEC Structure and Competence:

Redistribute the seats of AECs to always include a veterinarian, an ethologist, and an ethicist to ensure the wide yet specialized competence necessary to always be able to consider the animals’ needs and experiences and for the adequate promotion of in-depth ethical dialogue. Ensure that researchers in the AECs represent different research fields with a shared knowledge of a wide array of research methods, animal species (including wild ones and those commonly privately owned), and animal-free alternatives. This competence could be prioritized within each AEC or divided between different AECs [31]. If the latter approach is used, applications should be allocated accordingly to the AEC best suited to review them.

### 5.4. Mandatory Education and Training:

Provide written guidelines and mandatory continuous education for researchers and AEC members, for example in the shape of centralized educational programs managed by a governmental body such as the SBA, by the company or institution researchers are employed by, or ideally, a combination of the two. To ensure that each application undergoes a thorough ethical reflection beyond just technical refinements, the education should equip researchers and AEC members with a solid understanding of ethical reasoning related to animal research, emphasizing the steps involved in an ethical review. Important areas to highlight include the following: estimating benefits by considering research quality and the likelihood of success given the proposed research design; differentiating between types of benefits, such as knowledge versus practical outcomes; distinguishing between the research purpose and benefits; understanding and considering cumulative harm and recognizing how good animal welfare supports scientific quality; and applying and evaluating all three of the Rs. To manifest a proper understanding of the importance and difficulties of performing a proper ethical review for all projects, AEC members should also receive regular training in ethical evaluation, i.e., on how to assess different kinds of applications and on how to actively engage in ethical discussions. Additionally, if sufficient shared knowledge of different research methods, animal species, and alternative methods cannot be attained through the selection of members within an AEC, education and training to fill such knowledge gaps should be provided.

### 5.5. Support and Resources:

Ensure that 3Rs strategies are promoted and prioritized within research teams and institutions and that researchers have access to external support and resources from animal welfare bodies, national 3R centers, and similar agencies. This will help researchers better evaluate the need for using animals in the first place (Replacement), and when animal use is necessary, improve project design to better accommodate the specific needs and behaviors of different animal species and individuals, while effectively and confidently implementing the principles of Reduction and Refinement in their research. It will also likely improve information pertaining to the 3Rs provided in applications, ultimately facilitating the task of the AECs in assessing 3R compliance. Ensure corresponding support and resources for AECs so that they can accurately assess the suitability of housing systems, handling methods, etc., together with implementation of the 3Rs for the projects they review.

### 5.6. Empower AECs to Reject or Postpone Inadequate Applications:

Clarify the legal obligation that AECs question, reject, or postpone applications of inadequate quality, that are too comprehensive to be understood, or which are missing important information necessary for the performance of a proper ethical review.

### 5.7. Assess Compliance Through Random Audits:

Introduce random audits of approved applications conducted by the Swedish National Board on research animal ethics, to gain a better understanding of the quality and legal compliance of the ethical approval process and its compliance with current legislation. These audits would help improve and standardize the work of the AECs.

### 5.8. EU Commission Involvement:

Advocate for an EU Commission investigation into the implementation and adherence of Directive 2010/63/EU by member states to identify potential common struggles and successes.

### 5.9. Qualitative Research on Perception and Implementation:

Conduct qualitative research with applying researchers and AEC members, such as surveys or interviews, to understand how different parties involved in the ethical review perceive their tasks, responsibilities, and the ethical approval as a whole. Based on this, future revisions of the ethical review may be tailored to address identified specific needs to improve the overall effectiveness of the process.

By addressing the identified gaps and challenges as suggested, we see great potential that the impartiality, consistency, transparency, and effectiveness of the Swedish ethical review process would increase. This would not only ensure that animal use in research complies better with both national and EU regulations, but also that the animals used are more adequately considered in the ethical equation. Finally, it would enable trust amongst the public that animal research abides by set standards, not only in theory but, more importantly, in practice.

## 6. Conclusions

In conclusion, our study has shed light on regulatory ambiguities and practical challenges that impact the critical function that is the ethical review of animal research in Sweden. Our study, grounded in a detailed analysis of regulatory frameworks and the careful scrutiny of submitted applications for ethical review, highlights significant challenges faced in ensuring the ethical integrity of animal research.

Several of these challenges, we believe, may be directly associated with incertitude related to interpreting and abiding by relevant regulations. In determining what constitutes “relevant” information for ethical review, there is a clear disconnect between the expectations set by Directive 2010/63/EU and the intentional flexibility of the L150 as explained by the SBA. This ambiguity, although sprung from good intentions aiming to provide AECs with discretion to adapt to the specifics of each project proposal, risks diluting the core principles of Directive 2010/63/EU and undermining the principles of impartiality, consistency, and transparency foundational to ethical review processes and the broader Rule of Law. In essence, these potential consequences pinpoint the inherent intricacy of harmonizing legislation across EU member states while allowing for subjective interpretation by AECs.

Our study has further showed that applications often lack clarity and completeness concerning basic facts about the animals and projects, as well as regarding descriptions of harm to the animals and purposes and benefits of the research. We believe this may be, at least in part, due to discrepancies between the application form of 2020 and the L150, resulting in additional confusion about what information to include or not. Undoubtedly, not being in possession of all relevant details complicates the AECs’ ability to perform thorough HBAs and to evaluate researchers’ applications of the 3Rs effectively, increasing the risk of compromised animal welfare. For this reason, ensuring that sufficient information about the animals’ situation is provided for each proposed project submitted for ethical review should be a top priority.

## Figures and Tables

**Figure 1 animals-15-02771-f001:**
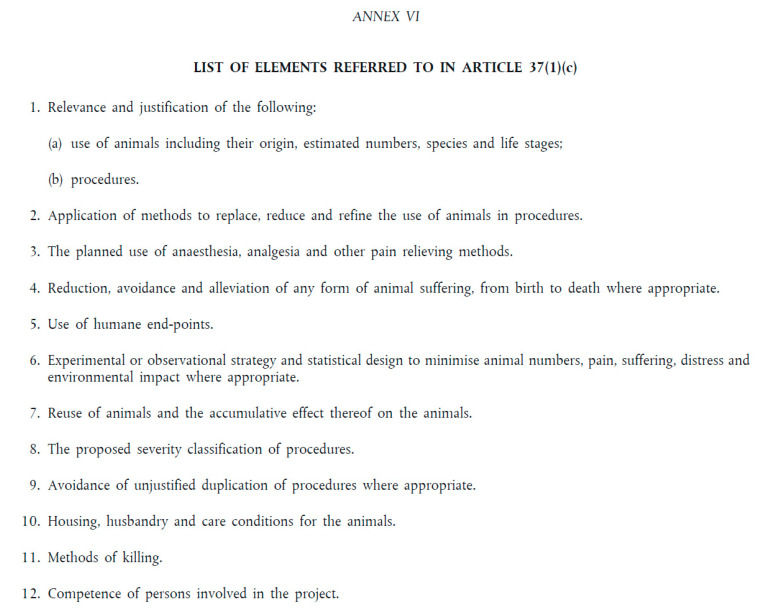
Excerpt from Directive 2010/63/EU showing the list of elements in Annex VI to be provided within the application for ethical approval according to Article 37.

**Figure 2 animals-15-02771-f002:**
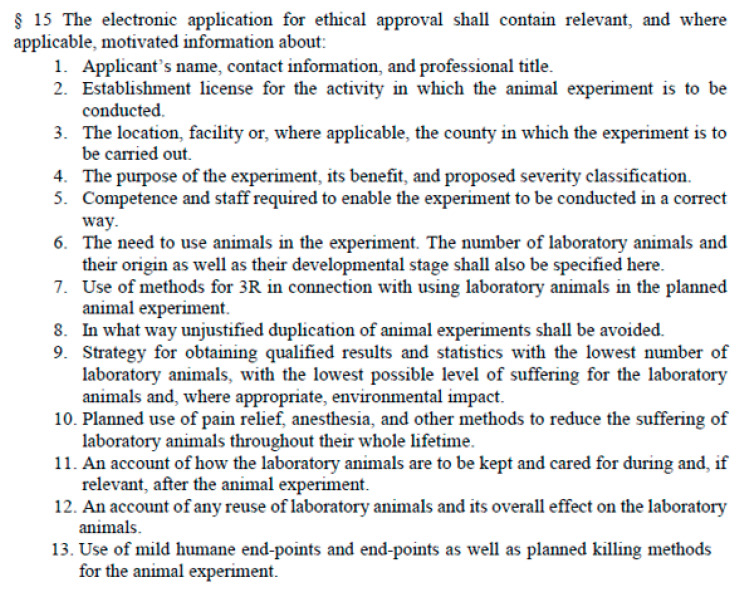
Excerpt from the English version of The Swedish Board of Agriculture’s Regulations and General Advice on Laboratory Animals (SJVFS 2019:09) showing the list of elements to be provided within the application for ethical approval according to Chapter 2 Section 15. Please note that in this version, the word “before” has been lost in translation under point 11. To mirror the Swedish original document, the wording of this sentence should be “An account of how the laboratory animals are to be kept and cared for *before*, during and, if relevant, after the animal experiment.” (our emphasis).

**Figure 3 animals-15-02771-f003:**
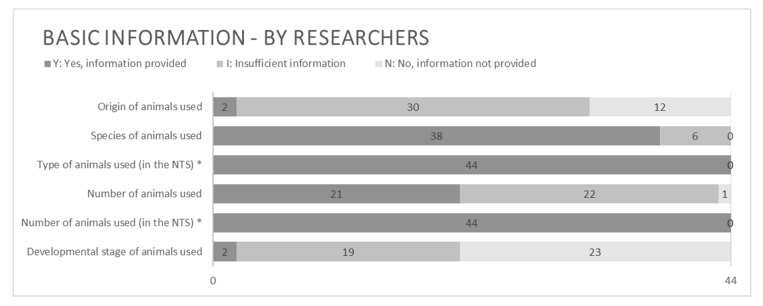
Basic information regarding animal use provided by applying researchers. N = 44. * For information provided regarding type and number of animals used (within the NTS), only the grades Y or N could be obtained. See the guide in the Appendix A for details.

**Figure 4 animals-15-02771-f004:**
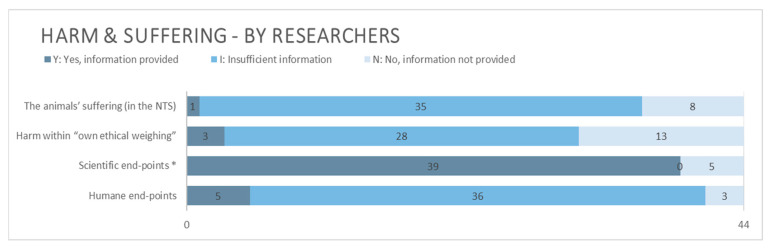
Information related to harm and suffering provided by applying researchers. N = 44. * For information provided regarding scientific end-points, only the grades Y or N could be obtained. See the guide in the Appendix A for details.

**Figure 5 animals-15-02771-f005:**
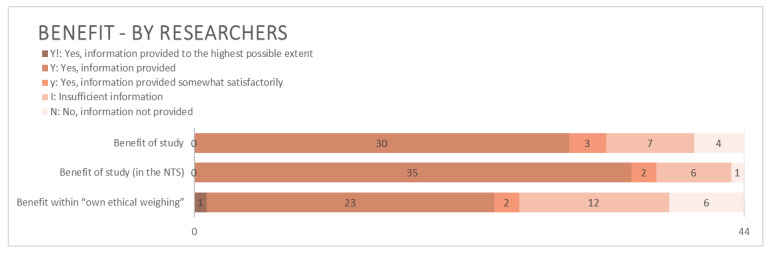
Information related to benefit provided by applying researchers. N = 44.

**Figure 6 animals-15-02771-f006:**
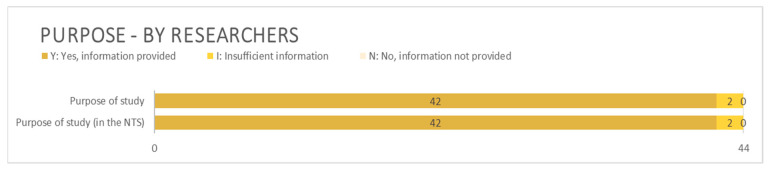
Information related to purpose provided by applying researchers. N = 44.

**Figure 7 animals-15-02771-f007:**
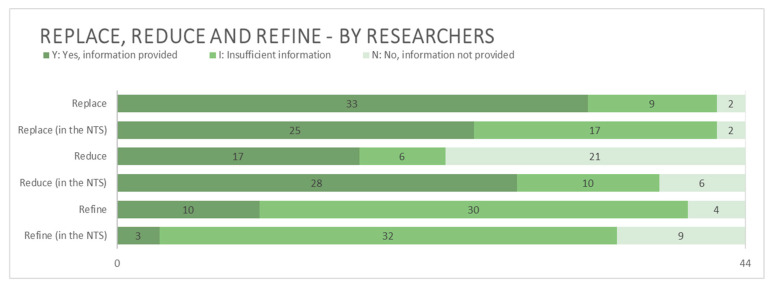
Information related to Replace, Reduce, and Refine provided by applying researchers. N = 44.

## Data Availability

Applications for ethical review and their corresponding decisions are publicly accessible and available in full upon request from the Swedish district courts (contact details: https://jordbruksverket.se/djur/ovriga-djur/forsoksdjur-och-djurforsok/tillstand-och-godkannanden, accessed on 27 August 2025). The summarized datasets generated and analyzed during the current study may be made available from the corresponding author upon reasonable request.

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
