# Peer review of "Approved Ambiguities: An Analysis of Applications for the Ethical Review of Animal Research in Sweden—Focusing on Harm, Benefit, and the 3Rs"

_animals, 2025, doi:10.3390/ani15192771_

Round 1

Reviewer 1 Report

Comments and Suggestions for Authors

The manuscript was reviewed. It addresses an important topic, but its primary scientific flaw is the exclusive focus on application content without sufficiently analyzing the decision-making process of Animal Ethics Committees (AECs). This limits the understanding of the practical impact of incomplete information on ethical reviews, reducing the article’s comprehensiveness and impact due to insufficient evidence supporting claims about deficiencies in harm-benefit analysis (HBA) or the 3Rs.

In its current form, the manuscript is not suitable for direct acceptance due to this limitation. However, given the topic’s significance, robust methodology, and practical recommendations, it has strong potential for acceptance after major revisions. The authors are requested to:

  1. Add qualitative data: Conduct interviews with 10–15 AEC members from various Swedish committees or analyze plenary meeting minutes to show how committees handle incomplete information.
  2. Include analysis of final decisions: Examine how incomplete applications are approved, rejected, or deferred, using publicly available Swedish court documents (e.g., lines 190–191).
  3. Adopt a mixed-methods approach: Combine quantitative content analysis with qualitative data to strengthen the study.

These revisions will address gaps and enhance the article’s conclusions.

Thank you,

Reviewer 2 Report

Comments and Suggestions for Authors

Thank you for this nice paper.

I appreciate the methods used and the work that you have put into the content analysis. It is an important area for research.

I have a few points for clarification:

  • Please could you give more details on how this paper differs from your previous paper? There are references to methodological developments and regulatory amendment, but it would be good to list these in detail so that comparison can be made.
  • I found it difficult to understand what you meant by ‘motivation’ which is used in several different places. I think it has been translated from the original Swedish, but I wonder if it is used instead of ‘justification’, or ‘rationale’ and whether either of these terms may be better. ‘Motivation’ is used when talking about additional information provided by the applicant (3.1.2), but also when talking about reasons for the decisions made by the AEC (Motivation – supplementary materials). In UK/US English, motivation has a different meaning (incentive, stimulus). I found a South African interpretation that used motivation as ‘a set of facts and arguments used in support of a proposal’ but that is not in common use in UK/US. Indeed, legal motivation means the reason for someone’s actions or behaviour, so this could be confusing for the reader (although I noticed you used the same term in your previous paper).
  • Please simplify the legal methodology section (2.1.2) so that someone could replicate the method used. Currently, it moves between the recommended protocol and what you actually did – please rewrite in the active voice to facilitate replication.

I found a few minor typos and lexical errors, including the following:

  1. Please ensure that 3Rs is always in the plural form – it appears as ‘3R’ occasionally and it does not make sense in the singular form.
  2. Line 171 ‘These was made’ should be ‘These were made’
  3. Line 272 ‘either one could be deemed’ should be ‘any one could be deemed’ as you are referring to more than two points.
  4. Line 655 ‘since it’s achievement’ should be ‘since its achievement’

Reviewer 3 Report

Comments and Suggestions for Authors

GENERAL

This is a very well written and interesting article, that will surely find a wide readership.  Indeed, queries below regarding clarification are made because there will no doubt be readers who come from jurisdictions where the rules differ – eg each research institution has its own AEC.

Did this project itself, require ethics approval. In Australia, this would have required human research ethics approval (because the research involves how AECs make decisions). However, I notice that the material used is in the public domain – perhaps clarify with authors.

Line 180 indicates that diaries etc were requested from each AEC – were the AECs obliged to hand over these materials?  Lines 189-191 indicate that the material is publicly available. In some jurisdictions this is unusual. Readers will be interested in how this works. A couple of sentences to explain whether making material publicly available is unique to Sweden, or part of EU directive; and also, why the material is required to be publicly available (social licence arguments perhaps) and whether there are issues of confidentiality regarding the research, and how these are managed.

Lines 92-93 – could this be clarified a little more.. Eg how many members must an AEC have, what is the ration between scientists, animal welfarists and members of the public. What qualifications does the chair have. Why lawyers? This is also important to conclusions drawn by the researchers in lines 519-521, an 611-612 – is there no one who speaks for the animals?  And also to the suggestion for improvement lines 742-3.

97-105 – Sweden – this is interesting. It appears that research institutions do not have their own AECs. If this is the case, please explain a little more clearly.  Also is there any government oversight eg are there inspections by a government regulatory body.

Lines 108-109 – how are the applications forms devised and designed – ie who designs them. Also lines 687-688 seem to indicate that the form is centrally produced (if this is not correct, perhaps explain a bit more)

SPECIFIC

Lines 171 – if the editors are concerned there is a split infinitive

Lines 192-230 – data analysis, show that the project has been carefully thought out. (also evident in explanation of laws in part 3.1.1)

Line 390 “Refinement measures to serve as a meaningful benchmark” – may need an explanation –

Line 447-448 – If this is the case, is it not open to AEC to reject the application/or more likely, to request further information.

Lines 465-466 – agree that in field studies knowing the number of species to be observed is not possible. But, what is meant of “catching of wild fish” – was the research targeting a specific species, or was it research designed to identify abundance and type of species Do we know whether the fish were released or captured for experimentation – assuming this matters from the perspective or harm assessment and harm management.

Line 537-539 – housing. Are there directives or other rules relevant here?

Lines 666-668 – probably a bit too peripheral for this study, but a point that has always concerned me, is whether AECs are brave enough to refuse approval for projects that have already attracted funding. A project has gone through a review in the grant application process may sway AECs in that it seems worthy.  This point is sort of picked up in lines 788-90 – perhaps earlier set out what the powers of the AECs are.

Round 2

Reviewer 1 Report

Comments and Suggestions for Authors

OK